# Variational Memory Encoder-Decoder

**Hung Le, Truyen Tran, Thin Nguyen and Svetha Venkatesh**
Applied AI Institute, Deakin University, Geelong, Australia
{lethai,truyen.tran,thin.nguyen,svetha.venkatesh}@deakin.edu.au

## Abstract

Introducing variability while maintaining coherence is a core task in learning to generate utterances in conversation. Standard neural encoder-decoder models and their extensions using conditional variational autoencoder often result in either trivial or digressive responses. To overcome this, we explore a novel approach that injects variability into neural encoder-decoder via the use of external memory as a mixture model, namely Variational Memory Encoder-Decoder (VMED). By associating each memory read with a mode in the latent mixture distribution at each timestep, our model can capture the variability observed in sequential data such as natural conversations. We empirically compare the proposed model against other recent approaches on various conversational datasets. The results show that VMED consistently achieves significant improvement over others in both metric-based and qualitative evaluations.

## 1   Introduction

Recent advances in generative modeling have led to exploration of generative tasks. While generative models such as GAN [12] and VAE [19, 29] have been applied successfully for image generation, learning generative models for sequential discrete data is a long-standing problem. Early attempts to generate sequences using RNNs [13] and neural encoder-decoder models [17, 35] gave promising results, but the deterministic nature of these models proves to be inadequate in many realistic settings. Tasks such as translation, question-answering and dialog generation would benefit from stochastic models that can produce a variety of outputs for an input. For example, there are several ways to translate a sentence from one language to another, multiple answers to a question and multiple responses for an utterance in conversation.

Another line of research that has captured attention recently is memory augmented neural networks (MANNs). Such models have larger memory capacity and thus "remember" temporally distant information in the input sequence and provide a RAM-like mechanism to support model execution. MANNs have been successfully applied to long sequence prediction tasks [14, 33] demonstrating great improvement when compared to other recurrent models. However, the role of memory in sequence generation has not been well understood.

For tasks involving language understanding and production, handling intrinsic uncertainty and latent variations is necessary. The choice of words and grammars may change erratically depending on speaker intentions, moods and previous languages used. The underlying RNN in neural sequential models finds it hard to capture the dynamics and their outputs are often trivial or too generic [23]. One way to overcome these problems is to introduce variability into these models. Unfortunately, sequential data such as speech and natural language is a hard place to inject variability [30] since they require a coherence of grammars and semantics yet allow freedom of word choice.

We propose a novel hybrid approach that integrates MANN and VAE, called Variational Memory Encoder-Decoder (VMED), to model the sequential properties and inject variability in sequence generation tasks. We introduce latent random variables to model the variability observed in the data

and capture dependencies between the latent variables across timesteps. Our assumption is that there are latent variables governing an output at each timestep. In the conversation context, for instance, the latent space may represent the speaker's hidden intention and mood that dictate word choice and grammars. For a rich latent multimodal space, we use a Mixture of Gaussians (MoG) because a spoken word's latent intention and mood can come from different modes, e.g., whether the speaker is asking or answering, or she/he is happy or sad. By modeling the latent space as an MoG where each mode associates with some memory slot, we aim to capture multiple modes of the speaker's intention and mood when producing a word in the response. Since the decoder in our model has multiple read heads, the MoG can be computed directly from the content of chosen memory slots. Our external memory plays a role as a mixture model distribution generating the latent variables that are used to produce the output and take part in updating the memory for future generative steps.

To train our model, we adapt Stochastic Gradient Variational Bayes (SGVB) framework [19]. Instead of minimizing the $KL$ divergence directly, we resort to using its variational approximation [15] to accommodate the MoG in the latent space. We show that minimizing the approximation results in $KL$ divergence minimization. We further derive an upper bound on our total timestep-wise $KL$ divergence and demonstrate that minimizing the upper bound is equivalent to fitting a continuous function by a scaled MoG. We validate the proposed model on the task of conversational response generation. This task serves as a nice testbed for the model because an utterance in a conversation is conditioned on previous utterances, the intention and the mood of the speaker. Finally, we evaluate our model on two open-domain and two closed-domain conversational datasets. The results demonstrate our proposed VMED gains significant improvement over state-of-the-art alternatives.

## 2 Preliminaries

### 2.1 Memory-augmented Encoder-Decoder Architecture

A memory-augmented encoder-decoder (MAED) consists of two neural controllers linked via external memory. This is a natural extension to read-write MANNs to handle sequence-to-sequence problems. In MAED, the memory serves as a compressor that encodes the input sequence to its memory slots, capturing the most essential information. Then, a decoder will attend to these memory slots looking for the cues that help to predict the output sequence. MAED has recently demonstrated promising results in machine translation [5, 37] and healthcare [20, 21, 28]. In this paper, we advance a recent MAED known as DC-MANN described in [20] where the powerful DNC [14] is chosen as the external memory. In DNC, memory accesses and updates are executed via the controller's reading and writing heads at each timestep. Given current input $x_t$ and a set of $K$ previous read values from memory $r_{t-1} = \left[ r_{t-1}^1, r_{t-1}^2, ..., r_{t-1}^K \right]$, the controllers compute read-weight vector $w_t^{i,r}$ and write-weight vector $w_t^w$ for addressing the memory $M_t$. There are two versions of decoding in DC-MANN: write-protected and writable memory. We prefer to allow writing to the memory during inference because in this work, we focus on generating diverse output sequences, which requires a dynamic memory for both encoding and decoding process.

### 2.2 Conditional Variational Autoencoder (CVAE) for Conversation Generation

A dyadic conversation can be represented via three random variables: the conversation context $x$ (all the chat before the response utterance), the response utterance $y$ and a latent variable $z$, which is used to capture the latent distribution over the reasonable responses. A variational autoencoder conditioned on $x$ (CVAE) is trained to maximize the conditional log likelihood of $y$ given $x$, which involves an intractable marginalization over the latent variable $z$, i.e.,:

$$p\left(y \mid x\right) = \int_z p\left(y, z \mid x\right) dz = \int_z p\left(y \mid x, z\right) p\left(z \mid x\right) dz \tag{1}$$

Fortunately, CVAE can be efficiently trained with the Stochastic Gradient Variational Bayes (SGVB) framework [19] by maximizing the variational lower bound of the conditional log likelihood. In a typical CVAE work, $z$ is assumed to follow multivariate Gaussian distribution with a diagonal covariance matrix, which is conditioned on $x$ as $p_\phi\left(z \mid x\right)$ and a recognition network $q_\theta(z \mid x, y)$ to approximate the true posterior distribution $p(z \mid x, y)$. The variational lower bound becomes:

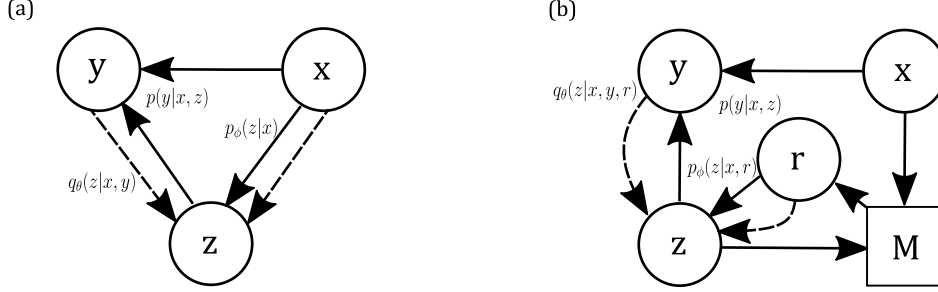

Figure 1: Graphical Models of the vanilla CVAE (a) and our proposed VMED (b)

$$L\left(\phi,\theta;y,x\right)=-KL\left(q_{\theta}\left(z\mid x,y\right)\parallel p_{\phi}\left(z\mid x\right)\right)+\mathbb{E}_{q_{\theta}(z|x,y)}\left[\log p\left(y\mid x,z\right)\right]\leq\log p\left(y\mid x\right) \quad (2)$$

With the introduction of the neural approximator $q_{\theta}(z\mid x,y)$ and the reparameterization trick [18], we can apply the standard back-propagation to compute the gradient of the variational lower bound. Fig. 1(a) depicts elements of the graphical model for this approach in the case of using CVAE.

## 3   Methods

Built upon CVAE and partly inspired by VRNN [8], we introduce a novel memory-augmented variational recurrent network dubbed Variational Memory Encoder-Decoder (VMED). With an external memory module, VMED explicitly models the dependencies between latent random variables across subsequent timesteps. However, unlike the VRNN which uses hidden values of RNN to model the latent distribution as a Gaussian, our VMED uses read values $r$ from an external memory $M$ as a Mixture of Gaussians (MoG) to model the latent space. This choice of MoG also leads to new formulation for the prior $p_{\phi}$ and the posterior $q_{\theta}$ mentioned in Eq. (2). The graphical representation of our model is shown in Fig. 1(b).

### 3.1   Generative Process

The VMED includes a CVAE at each time step of the decoder. These CVAEs are conditioned on the context sequence via $K$ read values $r_{t-1} = \left[r_{t-1}^{1}, r_{t-1}^{2}, ..., r_{t-1}^{K}\right]$ from the external memory. Since the read values are conditioned on the previous state of the decoder $h_{t-1}^{d}$, our model takes into account the temporal structure of the output. Unlike other designs of CVAE where there is often only one CVAE with a Gaussian prior for the whole decoding process, our model keeps reading the external memory to produce the prior as a Mixture of Gaussians at every timestep. At the $t$-th step of generating an utterance in the output sequence, the decoder will read from the memory $K$ read values, representing $K$ modes of the MoG. This multi-modal prior reflects the fact that given a context $x$, there are different modes of uttering the output word $y_t$, which a single mode cannot fully capture. The MoG prior distribution is modeled as:

$$g_{t} = p_{\phi}\left(z_{t}\mid x, r_{t-1}\right) = \sum_{i=1}^{K} \pi_{t}^{i,x}\left(x, r_{t-1}^{i}\right)\mathcal{N}\left(z_{t}; \mu_{t}^{i,x}\left(x, r_{t-1}^{i}\right), \sigma_{t}^{i,x}\left(x, r_{t-1}^{i}\right)^{2}\mathbf{I}\right) \quad (3)$$

We treat the mean $\mu_{t}^{i,x}$ and standard deviation (s.d.) $\sigma_{t}^{i,x}$ of each Gaussian distribution in the prior as neural functions of the context sequence $x$ and read vectors from the memory. The context is encoded into the memory by an $LSTM^{E}$ encoder. In decoding, the decoder $LSTM^{D}$ attends to the memory and choose $K$ read vectors. We split each read vector into two parts $r^{i,\mu}$ and $r^{i,\sigma}$, each of which is used to compute the mean and s.d., respectively: $\mu_{t}^{i,x} = r_{t-1}^{i,\mu}$, $\sigma_{t}^{i,x} = softplus\left(r_{t-1}^{i,\sigma}\right)$. Here we use the softplus function for computing s.d. to ensure the positiveness. The mode weight $\pi_{t}^{i,x}$ is chosen based on the read attention weights $w_{t-1}^{i,r}$ over memory slots. Since we use soft-attention, a read value is computed from all slots yet the main contribution comes from the one

**Algorithm 1** VMED Generation

---

**Require:** Given $p_\phi$, $[r_0^1, r_0^2, ..., r_0^K]$, $h_0^d$, $y_0^*$
1: **for** $t = 1, T$ **do**
2:      Sampling $z_t \sim p_\phi(z_t \mid x, r_{t-1})$ in Eq. (3)
3:      Compute: $o_t^d, h_t^d = LSTM^D\left([y_{t-1}^*, z_t], h_{t-1}^d\right)$
4:      Compute the conditional distribution: $p(y_t \mid x, z_{\leq t}) = softmax\left(W_{out} o_t^d\right)$
5:      Update memory and read $[r_t^1, r_t^2, ..., r_t^K]$ using $h_t^d$ as in DNC
6:      Generate output $y_t^* = \underset{y \in Vocab}{argmax}\, p(y_t = y \mid x, z_{\leq t})$
7: **end for**

---

with highest attention score. Thus, we pick the maximum attention score in each read weight and normalize to become the mode weights: $\pi_t^{i,x} = \max w_{t-1}^{i,r} / \sum_{i=1}^{i=K} \max w_{t-1}^{i,r}$.

Armed with the prior, we follow a recurrent generative process by alternatively using the memory to compute the MoG and using latent variable $z$ sampled from the MoG to update the memory and produce the output conditional distribution. The pseudo-algorithm of the generative process is given in Algorithm 1.

### 3.2 Neural Posterior Approximation

At each step of the decoder, the true posterior $p(z_t \mid x, y)$ will be approximated by a neural function of $x$, $y$ and $r_{t-1}$, denoted as $q_\theta(z_t \mid x, y, r_{t-1})$. Here, we use a Gaussian distribution to approximate the posterior. The unimodal posterior is chosen because given a response $y$, it is reasonable to assume only one mode of latent space is responsible for this response. Also, choosing a unimodel will allow the reparameterization trick during training and reduce the complexity of $KL$ divergence computation. The approximated posterior is computed by the following the equation:

$$f_t = q_\theta(z_t \mid x, y_{\leq t}, r_{t-1}) = \mathcal{N}\left(z_t; \mu_t^{x,y}(x, y_{\leq t}, r_{t-1}), \sigma_t^{x,y}(x, y_{\leq t}, r_{t-1})^2 \mathbf{I}\right) \quad (4)$$

with mean $\mu_t^{x,y}$ and s.d. $\sigma_t^{x,y}$. We use an $LSTM^U$ utterance encoder to model the ground truth utterance sequence up to timestep $t$-th $y_{\leq t}$. The $t$-th hidden value of the $LSTM^U$ is used to represent the given data in the posterior: $h_t^u = LSTM^U\left(y_t, h_{t-1}^u\right)$. The neural posterior combines the read values $\mathbf{r}_t = \sum_{i=1}^{K} \pi_t^{i,x} r_{t-1}^i$ together with the ground truth data to produce the Gaussian posterior: $\mu_t^{x,y} = W_\mu[\mathbf{r}_t, h_t^u]$, $\sigma_t^{x,y} = softplus(W_\sigma[\mathbf{r}_t, h_t^u])$. In these equations, we use learnable matrix weights $W_\mu$ and $W_\sigma$ as a recognition network to compute the mean and s.d. of the posterior, ensuring that the distribution has the same dimension as the prior. We apply the reparamterization trick to calculate the random variable sampled from the posterior as $z_t' = \mu_t^{x,y} + \sigma_t^{x,y} \odot \epsilon$, $\epsilon \in \mathcal{N}(0, \mathbf{I})$. Intuitively, the reparameterization trick bridges the gap between the generation model and the inference model during the training.

### 3.3 Learning

In the training phase, the neural posterior is used to produce the latent variable $z_t'$. The read values from memory are used directly as the MoG priors and the priors are trained to approximate the posterior by reducing the $KL$ divergence. During testing, the decoder uses the prior for generating latent variable $z_t$, from which the output is computed. The training and testing diagram is illustrated in Fig. 2. The objective function becomes a timestep-wise variational lower bound by following similar derivation presented in [8]:

$$\mathcal{L}(\theta, \phi; y, x) = E_{q*}\left[\sum_{t=1}^{T} -KL\left(q_\theta(z_t \mid x, y_{\leq t}, r_{t-1}) \| p_\phi(z_t \mid x, r_{t-1})\right) + \log p(y_t \mid x, z_{\leq t})\right]$$
$$(5)$$

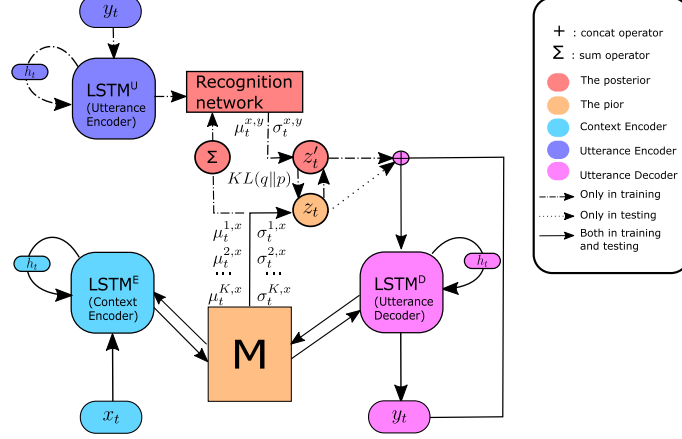

Figure 2: Training and testing of VMED

where $q* = q_\theta\left(z_{\leq T} \mid x, y_{\leq T}, r_{<T}\right)$. To maximize the objective function, we have to compute $KL$ divergence between $f_t = q_\theta\left(z_t \mid x, y_{\leq t}, r_{t-1}\right)$ and $g_t = p_\phi\left(z_t \mid x, r_{t-1}\right)$. Since there is no closed-form for this $KL\left(f_t \parallel g_t\right)$ between Gaussian $f_t$ and Mixture of Gaussians $g_t$, we use a closed-form approximation named $D_{var}$ [15] to replace the $KL$ term in the objective function. For our case: $KL\left(f_t \parallel g_t\right) \approx D_{var}\left(f_t \parallel g_t\right) = -\log \sum_{i=1}^{K} \pi^i e^{-KL\left(f_t \parallel g_t^i\right)}$. Here, $KL\left(f_t \parallel g_t^i\right)$ is the $KL$ divergence between two Gaussians and $\pi^i$ is the mode weight of $g_t$. The final objective function is:

$$
\begin{aligned}
\mathcal{L} = &\sum_{t=1}^{T} \log \sum_{i=1}^{K} \left[\pi_t^{i,x} \exp\left(-KL\left(\mathcal{N}\left(\mu_t^{x,y}, \sigma_t^{x,y2}\mathbf{I}\right) \parallel \mathcal{N}\left(\mu_t^{i,x}, \sigma_t^{i,x2}\mathbf{I}\right)\right)\right)\right] \\
&+ \frac{1}{L} \sum_{t=1}^{T}\sum_{l=1}^{L} \log p\left(y_t \mid x, z_{\leq t}^{(l)}\right)
\end{aligned}
\tag{6}
$$

### 3.4 Theoretical Analysis

We now show that by modeling the prior as MoG and the posterior as Gaussian, minimizing the approximation results in $KL$ divergence minimization. Let define the log-likelihood $L_f\left(g\right) = E_{f(x)}\left[\log g\left(x\right)\right]$, we have (see Supplementary material for full derivation):

$$
\begin{aligned}
L_f\left(g\right) \geq &\log \sum_{i=1}^{K} \pi^i e^{-KL\left(f \parallel g^i\right)} + L_f\left(f\right) = -D_{var} + L_f\left(f\right) \\
\Rightarrow D_{var} \geq &L_f\left(f\right) - L_f\left(g\right) = KL\left(f \parallel g\right)
\end{aligned}
$$

Thus, minimizing $D_{var}$ results in $KL$ divergence minimization. Next, we establish an upper bound on the total timestep-wise $KL$ divergence in Eq. (5) and show that minimizing this upper bound is equivalent to fitting a continuous function by a scaled MoG. The total timestep-wise $KL$ divergence reads:

$$
\sum_{t=1}^{T} KL\left(f_t \parallel g_t\right) = \int_{-\infty}^{+\infty} \sum_{t=1}^{T} f_t\left(x\right) \log\left[f_t\left(x\right)\right] dx \quad - \int_{-\infty}^{+\infty} \sum_{t=1}^{T} f_t\left(x\right) \log\left[g_t\left(x\right)\right] dx
$$

Table 1: BLEU-1, 4 and A-Glove on testing datasets. B1, B4, AG are acronyms for BLEU-1, BLEU-4, A-Glove metrics, respectively (higher is better).

| Model | Cornell Movies | | | OpenSubtitle | | | LJ users | | | Reddit comments | | |
|---|---|---|---|---|---|---|---|---|---|---|---|---|
| | B1 | B4 | AG | B1 | B4 | AG | B1 | B4 | AG | B1 | B4 | AG |
| Seq2Seq | 18.4 | 9.5 | 0.52 | 11.4 | 5.4 | 0.29 | 13.1 | 6.4 | 0.45 | 7.5 | 3.3 | 0.31 |
| Seq2Seq-att | 17.7 | 9.2 | 0.54 | 13.2 | 6.5 | 0.42 | 11.4 | 5.6 | 0.49 | 5.5 | 2.4 | 0.25 |
| DNC | 17.6 | 9.0 | 0.51 | 14.3 | 7.2 | 0.47 | 12.4 | 6.1 | 0.47 | 7.5 | 3.4 | 0.28 |
| CVAE | 16.5 | 8.5 | 0.56 | 13.5 | 6.6 | 0.45 | 12.2 | 6.0 | 0.48 | 5.3 | 2.8 | 0.39 |
| VLSTM | 18.6 | 9.7 | 0.59 | 16.4 | 8.1 | 0.43 | 11.5 | 5.6 | 0.46 | 6.9 | 3.1 | 0.27 |
| VMED (K=1) | 20.7 | 10.8 | 0.57 | 12.9 | 6.2 | 0.44 | 13.7 | 6.9 | 0.47 | 9.1 | 4.3 | 0.39 |
| VMED (K=2) | 22.3 | 11.9 | **0.64** | 15.3 | 8.8 | 0.49 | 15.4 | 7.9 | **0.51** | 9.2 | 4.4 | 0.38 |
| VMED (K=3) | 19.4 | 10.4 | 0.63 | **24.8** | **12.9** | **0.54** | **18.1** | **9.8** | 0.49 | **12.3** | **6.4** | **0.46** |
| VMED (K=4) | **23.1** | **12.3** | 0.61 | 17.9 | 9.3 | 0.52 | 14.4 | 7.5 | 0.47 | 8.6 | 4.6 | 0.41 |

where $g_t = \sum\limits_{i=1}^{K} \pi_t^i g_t^i$ and $g_t^i$ is the $i$-th Gaussian in the MoG at timestep $t$-th. If at each decoding step, minimizing $D_{var}$ results in adequate $KL$ divergence such that the prior is optimized close to the neural posterior, according to Chebyshev's sum inequality, we can derive an upper bound on the total timestep-wise $KL$ divergence as (see Supplementary Materials for full derivation):

$$\int\limits_{-\infty}^{+\infty} \sum_{t=1}^{T} f_t(x) \log [f_t(x)] \, dx \quad - \int\limits_{-\infty}^{+\infty} \frac{1}{T} \sum_{t=1}^{T} f_t(x) \log \left[ \prod_{t=1}^{T} g_t(x) \right] dx \tag{7}$$

The left term is sum of the entropies of $f_t(x)$, which does not depend on the training parameter $\phi$ used to compute $g_t$, so we can ignore that. Thus given $f$, minimizing the upper bound of the total timestep-wise $KL$ divergence is equivalent to maximizing the right term of Eq. (7). Since $g_t$ is an MoG and products of MoG is proportional to an MoG, $\prod\limits_{t=1}^{T} g_t(x)$ is a scaled MoG (see Supplementary material for full proof). Maximizing the right term is equivalent to fitting function $\sum\limits_{t=1}^{T} f_t(x)$, which is sum of Gaussians and thus continuous, by a scaled MoG. This, in theory, is possible regardless of the form of $f_t$ since MoG is a universal approximator [1, 25].

## 4 Results

**Datasets and pre-processing**: We perform experiments on two collections: The first collection includes open-domain movie transcript datasets containing casual conversations: Cornell Movies[1] and OpenSubtitle[2]. They have been used commonly in evaluating conversational agents [24, 35]. The second are closed-domain datasets crawled from specific domains, which are question-answering of LiveJournal (LJ) users and Reddit comments on movie topics. For each dataset, we use 10,000 conversations for validating and 10,000 for testing.

**Baselines, implementations and metrics**: We compare our model with three deterministic baselines: the encoder-decoder neural conversational model (Seq2Seq) similar to [35] and its two variants equipped with attention mechanism [2] (Seq2Seq-att) and a DNC external memory [14] (DNC). The vanilla CVAE is also included in the baselines. To build this CVAE, we follow similar architecture introduced in [40] without bag-of-word loss and dialog act features[3]. A variational recurrent model without memory is also included in the baselines. The model termed VLSTM is implemented

Table 2: Examples of context-response pairs. /*/ denotes separations between stochastic responses.

| Input context | Response |
|---|---|
| **Reddit comment:** What is your favorite scene in film history ? Mine is the restaurant scene in the Godfather. | **Seq2Seq:** The scene in<br>**Seq2Seq-att:** The final<br>**DNC:** The scene in<br>**CVAE:** Inception god! Not by a shark /*/ Amour great /*/ Pro thing you know 3 dead<br>**VLSTM:** The scene in /*/ The of a dead /*/ The sky in scene<br>**VMED (K=3):** The opening scene from history movie /*/ The scene in a shot nights! Robin movie /*/ The psycho scene in fight from |
| **Reddit comment:** What actor will win an Oscar in the next 10 years ? | **Seq2Seq:** Colin<br>**Seq2Seq-att:** Liam Neeson<br>**DNC:** Tom Gyllenhaal<br>**CVAE:** Daryl and Aaron /*/ Carefully count Alfred Deniro /*/ Ponyo Joker possible<br>**VLSTM:** Michael Bullock /*/ Michael /*/ Michael De<br>**VMED (K=3):** Edward or Leo Dicaprio goes on /*/ Dicaprio will /*/ Dicaprio Tom has actually in jack on road |
| **LJ user:** Your very pretty and your is very cute. May i add ya to my friends list ? | **Seq2Seq:** I! Add and too you back<br>**Seq2Seq-att:** I did may! Away back<br>**DNC:** Sure you added<br>**CVAE:** Sure ! I can add you /*/ Thank cool of surely /*/ Thank hi of sure!<br>**VLSTM:** Haha of do go /*/ Haha of sure! Yes lol /*/ Haha sure! Go<br>**VMED (K=3):** Thank cool for sure /*/ Add definitely! Sure surely friends /*/ Thank girl go ahead |
| **Cornell movies:** Nobody can murder someone with their feelings . I mean that is crazy ? | **Seq2Seq:** Yes you are<br>**Seq2Seq-att:** Really not is it<br>**DNC:** Managing the truth<br>**CVAE:** Sure not to be in here /*/ Oh yes but i know! /*/ That to doing with here and<br>**VLSTM:** I am dead . ! That is ... /*/ I did not what way . /*/ I am not . But his things with ...<br>**VMED (K=4):** You are right! /*/ That is crazy /*/ You can't know Jimmy |
| **OpenSubtitle:** I'm obliged for your hospitality. I appreciate it and your husband too. | **Seq2Seq:** That is have got coming about these<br>**Seq2Seq-att:** May you not what nothing nobody<br>**DNC:** Yes i am taking this<br>**CVAE:** No . that for good! And okay /*/ All in the of two thing /*/ Sure. Is this!<br>**VLSTM:** I ... /*/ I understand /*/ I ! . ...<br>**VMED (K=3):** I know. I can afford /*/ I know nothing to store for you pass /*/ I know. Doing anymore you father |

based on LSTM instead of RNN as in VRNN framework [8]. We try our model VMED[4] with different number of modes ($K = 1, 2, 3, 4$). It should be noted that, when $K = 1$, our model's prior is exactly a Gaussian and the $KL$ term in Eq. (6) is no more an approximation. Details of dataset descriptions and model implementations are included in Supplementary material.

We report results using two performance metrics in order to evaluate the system from various linguistic points of view: (i) Smoothed Sentence-level BLEU [6]: BLEU is a popular metric that measures the geometric mean of modified ngram precision with a length penalty. We use BLEU-1 to 4 as our lexical similarity. (ii) Cosine Similarly of Sentence Embedding: a simple method to obtain sentence embedding is to take the average of all the word embeddings in the sentences [10]. We follow [40] and choose Glove [22] as the word embedding in measuring sentence similarly (A-Glove). To measure stochastic models, for each input, we generate output ten times. The metric between the ground truth and the generated output is calculated and taken average over ten responses.

**Metric-based Analysis**: We report results on four test datasets in Table 1. For BLEU scores, here we only list results for BLEU-1 and 4. Other BLEUs show similar pattern and will be listed in Supplementary material. As clearly seen, VMED models outperform other baselines over all metrics across four datasets. In general, the performance of Seq2Seq is comparable with other deterministic methods despite its simplicity. Surprisingly, CVAE or VLSTM does not show much advantage over deterministic models. As we shall see, although CVAE and VLSTM responses are diverse, they are often out of context. Among different modes of VMED, there is often one best fit with the data and thus shows superior performance. The optimal number of modes in our experiments often falls to $K = 3$, indicating that increasing modes does not mean to improve accuracy.

It should be noted that there is inconsistency between BLEU scores and A-Glove metrics. This is because BLEU measures lexicon matching while A-Glove evaluates semantic similarly in the embedding space. For examples, two sentences having different words may share the same meaning and lie close in the embedding space. In either case, compared to others, our optimal VMED always achieves better performance.

**Qualitative Analysis**

Table 2 represents responses generated by experimental models in reply to different input sentences. The replies listed are chosen randomly from 50 generated responses whose average of metric scores over all models are highest. For stochastic models, we generate three times for each input, resulting in three different responses. In general, the stochastic models often yield longer and diverse sequences as expected. For closed-domain cases, all models responses are fairly acceptable. Compared to the rest, our VMED's responds seem to relate more to the context and contain meaningful information. In this experiment, the open-domain input seems nosier and harder than the closed-domain ones, thus create a big challenge for all models. Despite that, the quality of VMED's responses is superior to others. Among deterministic models, DNC's generated responses look more reasonable than Seq2Seq's even though its BLEU scores are not always higher. Perhaps, the reference to external memory at every timestep enhances the coherence between output and input, making the response more related to the context. VMED may inherit this feature from its external memory and thus tends to produce reasonable responses. By contrast, although responses from CVAE and VLSTM are not trivial, they have more grammatical errors and sometimes unrelated to the topic.

# 5  Related Work

With the recent revival of recurrent neural networks (RNNs), there has been much effort spent on learning generative models of sequences. Early attempts include training RNN to generate the next output given previous sequence, demonstrating RNNs' ability to generate text and handwriting images [13]. Later, encoder-decoder architecture [34] enables generating a whole sequence in machine translation [17], text summation [27] and conversation generation [35]. Although these models have achieved significant empirical successes, they fall short to capture the complexity and variability of sequential processes.

These limitations have recently triggered a considerable effort on introducing variability into the encoder-decoder architecture. Most of the methods focus on conditional VAE (CVAE) by constructing a variational lower bound conditioned on the context. The setting can be found in many applications including machine translation [39] and dialog generation [4, 30, 31, 40]. A common trick is to place a neural net between the encoder and the decoder to compute the Gaussian prior and posterior of the CVAE. This design is further enhanced by the use of external memory [7] and reinforcement learning [38]. In contrast to this design, our VMED uses recurrent latent variable approach [8], that is, our model requires a CVAE for each step of generation. Besides, our external memory is used for producing the latent distribution, which is different from the one proposed in [7] where the memory is used only for holding long-term dependencies at sentence level. Compared to variational addressing scheme mentioned in [3], our memory uses deterministic addressing scheme, yet the memory content itself is used to introduce randomness to the architecture. More relevant to our work is GTMM [11] where memory read-outs involve in constructing the prior and posterior at every timesteps. However, this approach uses Gaussian prior without conditional context.

Using mixture of models instead of single Gaussian in VAE framework is not a new concept. Works in [9, 16] and [26] proposed replacing the Gaussian prior and posterior in VAE by MoGs for clustering and generating image problems. Works in [32] and [36] applied MoG prior to model transitions

between video frames and caption generation, respectively. These methods use simple feed forward network to produce Gaussian sub-distributions independently. In our model, on the contrary, memory slots are strongly correlated with each others, and thus modes in our MoG work together to define the shape of the latent distributions at specific timestep. To the best of our knowledge, our work is the first attempt to use an external memory to induce mixture models for sequence generation problems.

## 6 Conclusions

We propose a novel approach to sequence generation called Variational Memory Encoder-Decoder (VMED) that introduces variability into encoder-decoder architecture via the use of external memory as mixture model. By modeling the latent temporal dependencies across timesteps, our VMED produces a MoG representing the latent distribution. Each mode of the MoG associates with some memory slot and thus captures some aspect of context supporting generation process. To accommodate the MoG, we employ a $KL$ approximation and we demonstrate that minimizing this approximation is equivalent to minimizing the $KL$ divergence. We derive an upper bound on our total timestep-wise $KL$ divergence and indicate that the optimization of this upper bound is equivalent to fitting a continuous function by an scaled MoG, which is in theory possible regardless of the function form. This forms a theoretical basis for our model formulation using MoG prior for every step of generation. We apply our proposed model to conversation generation problem. The results demonstrate that VMED outperforms recent advances both quantitatively and qualitatively. Future explorations may involve implementing a dynamic number of modes that enable learning of the optimal $K$ for each timestep. Another aspect would be multi-person dialog setting, where our memory as mixture model may be useful to capture more complex modes of speaking in the dialog.

## Footnotes

[1] http://www.cs.cornell.edu/~cristian/Cornell_Movie-Dialogs_Corpus.html

[2] http://opus.nlpl.eu/OpenSubtitles.php

[3] Another variant of non-memory CVAE with MoG prior is also examined. We produce a set of MoG parameters by a feed forward network with the input as the last encoder hidden states. However, the model is hard to train and fails to converge with these datasets.

[4]Source code is available at `https://github.com/thaihungle/VMED`

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
