[Supplementary Material]

# Supplementary material for
# Variational Memory Encoder-Decoder

**Hung Le, Truyen Tran, Thin Nguyen and Svetha Venkatesh**
Applied AI Institute, Deakin University, Geelong, Australia
{lethai,truyen.tran,thin.nguyen,svetha.venkatesh}@deakin.edu.au

## 1   Derivation of the Upper Bound on the $KL$ divergence

**Theorem 1.** *The KL divergence between a Gaussian and a Mixture of Gaussians has an upper bound $D_{var}$.*

*Proof.* $D_{var}(f \parallel g)$ [3] is an approximation of $KL$ divergence between two Mixture of Gaussians (MoG), which is defined as the following:

$$D_{var}\left(f \parallel g\right) = \sum_{j} \pi_j^f \log \frac{\sum_{j'} \pi_{j'}^f e^{-KL\left(f_j \parallel f_{j'}\right)}}{\sum_{i} \pi_i^g e^{-KL\left(f_j \parallel g_i\right)}} \tag{1}$$

In our case, $f$ is a Gaussian, a special case of MoG where the number of mode equals one. Then, Eq. (1) becomes:

$$D_{var}\left(f \parallel g\right) = \log \frac{1}{\sum\limits_{i=1}^{K} \pi_i^g e^{-KL(f \parallel g^i)}} = -\log \sum_{i=1}^{K} \pi^i e^{-KL\left(f \parallel g^i\right)}$$

Let define the log-likelihood $L_f\left(g\right) = E_{f(x)}\left[\log g\left(x\right)\right]$, the lower bound for $L_f\left(g\right)$ can be also be derived, using variational parameters as follows:

$$
\begin{aligned}
L_f\left(g\right) &= E_f\left[\log\left(\sum_{i=1}^{K} \pi^i g^i\left(x\right)\right)\right] \\
&= \int\limits_{-\infty}^{+\infty} f\left(x\right) \log\left(\sum_{i=1}^{K} \beta^i \pi^i \frac{g^i\left(x\right)}{\beta^i}\right) dx \\
&\geq \sum_{i=1}^{K} \beta^i \int\limits_{-\infty}^{+\infty} f\left(x\right) \log\left(\pi^i \frac{g^i\left(x\right)}{\beta^i}\right) dx
\end{aligned}
$$

where $\beta^i \geq 0$ and $\sum\limits_{i=1}^{K} \beta^i = 1$. According to [2], maximizing the RHS of the above inequality with respect to $\beta^i$ provides a lower bound for $L_f\left(g\right)$:

$$L_f(g) \geq \log \sum_{i=1}^{K} \pi^i e^{-KL(f\|g^i)} + L_f(f)$$

$$= - D_{var} + L_f(f)$$

$$\Rightarrow D_{var} \geq L_f(f) - L_f(g)$$

$$= KL(f \parallel g)$$

Therefore, the $KL$ divergence has an upper bound: $D_{var}$. $\qquad\qquad\qquad\square$

## 2 Derivation of the Upper Bound on the Total Timestep-wise $KL$ Divergence

**Lemma 2.** *Chebyshev's sum inequality:*
*if*

$$a_1 \geq a_2 \geq ... \geq a_n$$

*and*

$$b_1 \geq b_2 \geq ... \geq b_n$$

then

$$\frac{1}{n} \sum_{k=1}^{n} a_k b_k \geq \left( \frac{1}{n} \sum_{k=1}^{n} a_k \right) \left( \frac{1}{n} \sum_{k=1}^{n} b_k \right)$$

*Proof.* Consider the sum:

$$S = \sum_{j=1}^{n} \sum_{k=1}^{n} (a_j - a_k)(b_j - b_k)$$

The two sequences are non-increasing, therefore $a_j - a_k$ and $b_j - b_k$ have the same sign for any $j, k$. Hence $S \geq 0$. Opening the brackets, we deduce:

$$0 \leq 2n \sum_{j=1}^{n} a_j b_j - 2 \sum_{j=1}^{n} a_j \sum_{k=1}^{n} b_k$$

whence:

$$\frac{1}{n} \sum_{j=1}^{n} a_j b_j \geq \left( \frac{1}{n} \sum_{j=1}^{n} a_j \right) \left( \frac{1}{n} \sum_{k=1}^{n} b_k \right)$$

$$\square$$

In our problem, $a_i = f_i(x)$ and $b_i = \log[g_i(x)]$, $i = \overline{1, T}$. Under the assumption that at each step, thanks to minimizing $D_{var}$, the approximation between the MoG and the Gaussian is adequate to preserve the order of these values, that is, if $f_i(x) \leq f_j(x)$, then $g_i(x) \leq g_j(x)$ and $\log[g_i(x)] \leq \log[g_j(x)]$. Without loss of generality, we hypothesize that $f_1(x) \leq f_2(x) \leq ... \leq f_T(x)$, then we have $\log[g_1(x)] \leq \log[g_2(x)] \leq ... \leq \log[g_T(x)]$. Thus, applying Lemma 2, we have:

$$\frac{1}{T} \sum_{t=1}^{T} f_t(x) \log[g_t(x)] \, dx \geq \frac{1}{T} \sum_{t=1}^{T} f_t(x) \frac{1}{T} \sum_{t=1}^{T} \log[g_t(x)] \, dx$$

$$\Rightarrow \int_{-\infty}^{+\infty} \sum_{t=1}^{T} f_t(x) \log\left[g_t(x)\right] dx \geq \int_{-\infty}^{+\infty} \frac{1}{T} \sum_{t=1}^{T} f_t(x) \sum_{t=1}^{T} \log\left[g_t(x)\right] dx$$

$$\Rightarrow \int_{-\infty}^{+\infty} \sum_{t=1}^{T} f_t(x) \log\left[g_t(x)\right] dx \geq \int_{-\infty}^{+\infty} \frac{1}{T} \sum_{t=1}^{T} f_t(x) \log\left[\prod_{t=1}^{T} g_t(x)\right] dx$$

Thus, the upper bound on the total timestep-wise $KL$ divergence reads:

$$\int_{-\infty}^{+\infty} \sum_{t=1}^{T} f_t(x) \log\left[f_t(x)\right] dx \quad - \int_{-\infty}^{+\infty} \frac{1}{T} \sum_{t=1}^{T} f_t(x) \log\left[\prod_{t=1}^{T} g_t(x)\right] dx$$

# 3 Proof $\prod_{t=1}^{T} g_t(x) = \prod_{t=1}^{T} \sum_{i=1}^{K} \pi_t^i g_t^i(x)$ is a Scaled MoG

**Lemma 3.** *Product of two Gaussians is a scaled Gaussian.*

*Proof.* Let $\mathcal{N}_x(\mu, \Sigma)$ denote a density of $x$, then

$$\mathcal{N}_x(\mu_1, \Sigma_1) \cdot \mathcal{N}_x(\mu_2, \Sigma_2) = c_c \mathcal{N}_x(\mu_c, \Sigma_c)$$

where:

$$c_c = \frac{1}{\sqrt{\det\left(2\pi\left(\Sigma_1 + \Sigma_2\right)\right)}} \exp\left(-\frac{1}{2}\left(m_1 - m_2\right)^T \left(\Sigma_1 + \Sigma_2\right)^{-1} \left(m_1 - m_2\right)\right)$$
$$m_c = \left(\Sigma_1^{-1} + \Sigma_2^{-1}\right)^{-1} \left(\Sigma_1^{-1} m_1 + \Sigma_2^{-1} m_2\right)$$
$$\Sigma_c = \left(\Sigma_1^{-1} + \Sigma_2^{-1}\right)$$

$\square$

**Lemma 4.** *Product of two MoGs is proportional to an MoG.*

*Proof.* Let $g_1(x) = \sum_{i=1}^{K_1} \pi_{1,i} \mathcal{N}_x(\mu_{1,i}, \Sigma_{1,i})$ and $g_2(x) = \sum_{j=1}^{K_2} \pi_{2,j} \mathcal{N}_x(\mu_{2,j}, \Sigma_{2,j})$ are two Mixtures of Gaussians. We have:

$$g_1(x) \cdot g_2(x) = \sum_{i=1}^{K_1} \pi_{1,i} \mathcal{N}_x(\mu_{1,i}, \Sigma_{1,i}) \cdot \sum_{j=1}^{K_2} \pi_{2,j} \mathcal{N}_x(\mu_{2,j}, \Sigma_{2,j})$$
$$= \sum_{i=1,j=1}^{K_1,K_2} \pi_{1,i} \pi_{2,j} \mathcal{N}_x(\mu_{1,i}, \Sigma_{1,i}) \cdot \mathcal{N}_x(\mu_{2,j}, \Sigma_{2,j}) \tag{2}$$

By applying Lemma 3 to Eq. (2), we have

$$g_1(x) \cdot g_2(x) = \sum_{i=1,j=1}^{K_1,K_2} \pi_{1,i} \pi_{2,j} c_{ij} \mathcal{N}_x(\mu_{ij}, \Sigma_{ij})$$
$$= C \sum_{i=1,j=1}^{K_1,K_2} \frac{\pi_{1,i} \pi_{2,j} c_{ij}}{C} \mathcal{N}_x(\mu_{ij}, \Sigma_{ij}) \tag{3}$$

where $C = \sum_{i=1,j=1}^{K_1,K_2} \pi_{1,i} \pi_{2,j} c_{ij}$. Clearly, Eq. (3) is proportional to an MoG with $K_1 \cdot K_2$ modes $\square$

**Theorem 5.** $\prod_{t=1}^{T} g_t(x) = \prod_{t=1}^{T} \sum_{i=1}^{K} \pi_t^i g_t^i(x)$ *is a scaled MoG.*

*Proof.* By induction from Lemma 4, we can easily show that product of $T$ MoGs is also proportional to an MoG. That means $\prod_{t=1}^{T} g_t(x)$ equals to a scaled MoG. □

## 4 Details of Data Descriptions and Model Implementations

Here we list all datasets used in our experiments:

- Open-domain datasets:

  - Cornell movie dialog: This corpus contains a large metadata-rich collection of fictional conversations extracted from 617 raw movies with 220,579 conversational exchanges between 10,292 pairs of movie characters. For each dialog, we preprocess the data by limiting the context length and the utterance output length to 20 and 10, respectively. The vocabulary is kept to top 20,000 frequently-used words in the dataset.

  - OpenSubtitles: This dataset consists of movie conversations in XML format. It also contains sentences uttered by characters in movies, yet it is much bigger and noisier than Cornell dataset. After preprocessing as above, there are more than 1.6 million pairs of contexts and utterance with chosen vocabulary of 40,000 words.

- Closed-domain datasets::

  - Live Journal (LJ) user question-answering dataset: question-answer dialog by LJ users who are members of anxiety, arthritis, asthma, autism, depression, diabetes, and obesity LJ communities[1]. After preprocessing as above, we get a dataset of more than 112,000 conversations. We limit the vocabulary size to 20,000 most common words.

  - Reddit comments dataset: This dataset consists of posts and comments about movies in Reddit website[2]. A single post may have multiple comments constituting a multi-people dialog amongst the poster and commentors, which makes this dataset the most challenging one. We crawl over four millions posts from Reddit website and after preprocessing by retaining conversations whose utterance's length are less than 20, we have a dataset of nearly 200 thousand conversations with a vocabulary of more than 16 thousand words.

We trained with the following hyperparameters (according to the performance on the validate dataset): word embedding has size 96 and is shared across everywhere. We initialize the word embedding from Google's Word2Vec [5] pretrained word vectors. The hidden dimension of LSTM in all controllers is set to 768 for all datasets except the big OpenSubtitles whose LSTM dimension is 1024. The number of LSTM layers for every controllers is set to 3. All the initial weights are sampled from a normal distribution with mean 0, standard deviation 0.1. The mini-batch size is chosen as 256. The models are trained end-to-end using the Adam optimizer [4] with a learning rate of 0.001 and gradient clipping at 10. For models using memory, we set the number and the size of memory slots to 16 and 64, respectively. As indicated in [1], it is not trivial to optimize VAE with RNN-like decoder due to the vanishing latent variable problem. Hence, to make the variational models in our experiments converge we have to use the $KL$ annealing trick by adding to the $KL$ loss term an annealing coefficient $\alpha$ starts with a very small value and gradually increase up to 1.

# 5 Full Reports on Model Performance

| Model | BLEU-1 | BLEU-2 | BLEU-3 | BLEU-4 | A-glove |
|-------|--------|--------|--------|--------|---------|
| Seq2Seq | 18.4 | 14.5 | 12.1 | 9.5 | 0.52 |
| Seq2Seq-att | 17.7 | 14.0 | 11.7 | 9.2 | 0.54 |
| DNC | 17.6 | 13.9 | 11.5 | 9.0 | 0.51 |
| CVAE | 16.5 | 13.0 | 10.9 | 8.5 | 0.56 |
| VLSTM | 18.6 | 14.8 | 12.4 | 9.7 | 0.59 |
| VMED (K=1) | 20.7 | 16.5 | 13.8 | 10.8 | 0.57 |
| VMED (K=2) | 22.3 | 18.0 | 15.2 | 11.9 | **0.64** |
| VMED (K=3) | 19.4 | 15.6 | 13.2 | 10.4 | 0.63 |
| VMED (K=4) | **23.1** | **18.5** | **15.5** | **12.3** | 0.61 |

Table 1: Results on Cornell Movies

| Model | BLEU-1 | BLEU-2 | BLEU-3 | BLEU-4 | A-glove |
|-------|--------|--------|--------|--------|---------|
| Seq2Seq | 11.4 | 8.7 | 7.1 | 5.4 | 0.29 |
| Seq2Seq-att | 13.2 | 10.2 | 8.4 | 6.5 | 0.42 |
| DNC | 14.3 | 11.2 | 9.3 | 7.2 | 0.47 |
| CVAE | 13.5 | 10.2 | 8.4 | 6.6 | 0.45 |
| VLSTM | 16.4 | 12.7 | 10.4 | 8.1 | 0.43 |
| VMED (K=1) | 12.9 | 9.5 | 7.5 | 6.2 | 0.44 |
| VMED (K=2) | 15.3 | 13.8 | 10.4 | 8.8 | 0.49 |
| VMED (K=3) | **24.8** | **19.7** | **16.4** | **12.9** | **0.54** |
| VMED (K=4) | 17.9 | 14.2 | 11.8 | 9.3 | 0.52 |

Table 2: Results on OpenSubtitles

| Model | BLEU-1 | BLEU-2 | BLEU-3 | BLEU-4 | A-glove |
|-------|--------|--------|--------|--------|---------|
| Seq2Seq | 13.1 | 10.1 | 8.3 | 6.4 | 0.45 |
| Seq2Seq-att | 11.4 | 8.7 | 7.1 | 5.6 | 0.49 |
| DNC | 12.4 | 9.6 | 7.8 | 6.1 | 0.47 |
| CVAE | 12.2 | 9.4 | 7.7 | 6.0 | 0.48 |
| VLSTM | 11.5 | 8.8 | 7.3 | 5.6 | 0.46 |
| VMED (K=1) | 13.7 | 10.7 | 8.9 | 6.9 | 0.47 |
| VMED (K=2) | 15.4 | 12.2 | 10.1 | 7.9 | **0.51** |
| VMED (K=3) | **18.1** | **14.8** | **12.4** | **9.8** | 0.49 |
| VMED (K=4) | 14.4 | 11.4 | 9.5 | 7.5 | 0.47 |

Table 3: Results on LJ users question-answering

| Model | BLEU-1 | BLEU-2 | BLEU-3 | BLEU-4 | A-glove |
|-------|--------|--------|--------|--------|---------|
| Seq2Seq | 7.5 | 5.5 | 4.4 | 3.3 | 0.31 |
| Seq2Seq-att | 5.5 | 4.0 | 3.1 | 2.4 | 0.25 |
| DNC | 7.5 | 5.6 | 4.5 | 3.4 | 0.28 |
| CVAE | 5.3 | 4.3 | 3.6 | 2.8 | 0.39 |
| VLSTM | 6.9 | 5.1 | 4.1 | 3.1 | 0.27 |
| VMED (K=1) | 9.1 | 6.8 | 5.5 | 4.3 | 0.39 |
| VMED (K=2) | 9.2 | 7.0 | 5.7 | 4.4 | 0.38 |
| VMED (K=3) | **12.3** | **9.7** | **8.1** | **6.4** | **0.46** |
| VMED (K=4) | 8.6 | 6.9 | 5.9 | 4.6 | 0.41 |

Table 4: Results on Reddit comments

## Footnotes

[1]https://www.livejournal.com/

[2]https://www.reddit.com/r/movies/