[Reviews · NeurIPS 2018]

Reviewer 1



The authors proposed a network composing of VAE and memory-augmented neural network (MANN), named Variational Memory Encoder-Decoder(VMED) to model sequential properties and inject variability at each step in sequence generation tasks. The authors use a Mixture of Gaussians (MoG) to represent the latent space, where each mode associates with some memory slots. The idea of training MoG prior, but a unimodal Gaussian posterior for each training data is nice. The authors have nice mathematical walk through. And, demonstrated better performance in BLEU-1, 4 and A-Glove scores in several different language modeling datasets. The example results in Table 2 are also quite impressive, showing diverse, high-quality sentences generated by the VMED model. The work is a step upon the variational RNN model (VRNN) (cited as [8] in the paper). I wonder why the authors did not try to run VRNN as one of the baseline models in Table 2. As discussed around L94-98, VRNN use hidden values of RNN to model the latent distribution as a Gaussian, and would be an important baseline to compare to VMED to see how much gain does the external memory provided beyond the LSTM-based model. While using the external memory in a VRNN setup is interesting. This paper, however, is not the first one suggesting this idea. Generative Temporal Models with Memory by Gemici et. al.,[1], is the first one I know and unfortunately, is missed from citation in this manuscript. Though, the methods in these two manuscripts are different in detail. (for example, using MoG prior in this work.) Also, the results in language modeling is quite impressive. [1] Gemici, M., Hung, C.C., Santoro, A., Wayne, G., Mohamed, S., Rezende, D.J., Amos, D., Lillicrap, T.: Generative Temporal Models with Memory. arXiv preprint arXiv:1702.04649 (2017)

Reviewer 2



This paper is well written. The proposed idea is sound. Introducing memory augmented encoder-decoder is interesting work.

Reviewer 3



Summary This paper introduces a new model for natural conversation modelling, the Variational Memory Encoder-Decoder (VMED). The purpose of the model is to learn a distribution over latent variables that captures time dependence and has different modes corresponding to for instance different moods and intentions during the conversation. In order to ensure coherence over time and variability of the generated conversations, an external memory is used as a mixture of Gaussians prior over the latent space, which is updated during the conversation. VMED outperforms all baselines in four conversational datasets. Originality As far as I understand the building blocks for this model include: 1. A Gaussian mixture model for the prior in the VAE. 2. The use of an external memory [5,35] to produce the parameters of the Gaussian mixture prior, which is updated at each time step, especially useful for coherence and multimodality in language tasks. 3. A probabilistic framework for sequence generation. [8], in the form of a conditional generative model [17, 38]. To my knowledge, all of these ideas are joined for the first time for sequence generation. Quality Besides being well motivated, the paper appears technically sound. The evaluation of the model as a whole is strong. The baselines are outperformed by the proposed model. However, it is hard to understand the influence of the different components on the increased performance over the baselines: i) the use for a multimodal prior distribution for a conditional VAE, ii) an external memory that adapts this distribution over time. It would provide useful insights to show experiments on this, for instance by comparing against a CVAE without an external memory but with a Gaussian mixture model prior. The details of the architectures used are limited. Clarity Overall this paper is written in an excellent manner. The formation of ideas is clear and the embedding in relevant literature is good. The figures are helpful in understanding the method, although I would encourage the authors to extend the captions to be more standalone. Significance The approach seems novel and relevant for language modelling to me. Since the main application in this paper is conversation modeling, I wonder what modeling choices need to be adapted in order to perform well on other sequence generation tasks, such as translation. Can the authors comment on this? Detailed comments/questions 1. In Eq. 2, There is a p missing inside the logarithm corresponding to the reconstruction term of the lower bound. 2. Should there be a solid arrow from x to z in figure 1? 3. Can the authors motivate why you choose to use a Gaussian mixture model as a prior instead of using for instance flows to create a multimodal distribution? 4. In Eq. 5, the minus sign in front of the sum should only be in front of the KL term. 5. In Eq. 6, the sum over i=1 to K is outside the log, whereas in the definition above for Dvar it is inside the log. Furthermore, it seems that z^l_t should be z^l_{<=t} in the last term. It would be good to explicitly mention that the last term is an MC estimate with samples from the posterior. Does r_{t-1} depends on z_